# ‘Are We Gonna Have to Pretend to Be a Straight Couple?’: Examining the Specific Detriment that Cisgenderism Places on Non-Binary Adoption and Fostering Applicants in the United Kingdom

**DOI:** 10.3390/bs14070614

**Published:** 2024-07-19

**Authors:** Claire Brown

**Affiliations:** School of Social Sciences, Humanities and Law, Teesside University, Middlesbrough TS1 3BX, UK; claire.brown@tees.ac.uk

**Keywords:** non-binary, trans, adoption, fostering, carer, parent, LGBTQ+

## Abstract

This exploratory qualitative study aimed to find out more about how the children’s social work system interacts with non-binary parents. It highlights the specific detriment that can be faced by non-binary people hoping to adopt or foster in the United Kingdom. Three key themes emerged: (1) Barriers for non-binary carers, (2) Prejudice in adoption and fostering matching processes and (3) Intersectional disadvantage. The study found that non-binary people experience specific detriment when endeavouring to start or grow their families, examining how cisgenderism operates to privilege some identities over others. Multiple barriers affect the way non-binary people try and navigate how their family lives in a society that is organised around binary gender identities. Cisgenderism can subtly and pervasively exert a devaluing of identities that sit outside of entrenched binary gender norms, influencing how people can understand and express their gender identities within society.

## 1. Introduction

For decades, the academy has deliberated the gendering processes of socialisation and norms associated with it; however, until recently such analysis has predominantly been in relation to a binary understanding of gender. This study adds the perspectives of non-binary people and their supporting social workers to the developing research base considering the diversity of gendered experiences within adoption and fostering in the United Kingdom. In-depth, semi-structured interviews were undertaken with six participants: four who define as non-binary or neutrois and have lived experience as carers or prospective carers, and two social workers with experience assessing and supporting non-binary carers. Braun and Clarke’s [1] six-stage thematic analysis method was employed, with an analysis framework integrating elements of cisgenderism, stigma theory, and Foucauldian analysis of discourse and power. 

A fundamental rationale for this research is that an entrenched binary understanding has arguably resulted in the silencing of people who identify outside of it or who defy its boundaries [2]. While the past two decades have produced a solid research base on the experiences of lesbian and gay carers [3,4,5,6], trans and non-binary voices have largely been unheeded or subsumed within wider research into lesbian, gay, bisexual, trans, and queer (LGBTQ+) communities [7,8].

This paper addresses the sparsity of research exploring how children’s social work in the United Kingdom engages with non-binary gender identities. Although non-binary people have attracted interest within mainstream media over recent years, such interest has not necessarily been productive in working towards the inclusion and acceptance of non-binary identities. Rather, the media has tended to situate non-binary people as ‘other’, compared to a normative conception of binary gender that it is assumed the wider public represents [9]. Stigma power is deployed via hidden forces such as unconscious bias that suppress gender diverse identities [10]. Increased focus on trans issues and rights within the media can then simultaneously serve to offer a platform for silenced voices to speak, whilst also positioning such voices as deviant. The result can then be a separating rather than a bringing together of trans and feminist communities [11,12,13].

Recent research suggests that the use of ‘non-binary’ as an umbrella option for identities that do not fit a man/woman binary categorical distinction has largely been accepted by the communities it represents [14,15]. As such, ‘non-binary’ is used within this paper to help readers make sense of the narratives; however, one participant identifies as neutrois, ‘which indicates dysphoria towards androgyny’ (Ash’s self-definition). ‘Trans’ is also used by participants as an umbrella term including a range of identities that describe people whose assigned gender at birth does not align with their own experience of gender [16]. Notwithstanding, this paper also strives to avoid homogenisation or delimitation of the terms ‘trans’ or ‘non-binary’, instead highlighting the importance of autonomy, self-identification, and the evolution of gender terminology [17].

## 2. Literature Review

Reflecting the narrative methodology of the study, a narrative literature review was conducted to weave together research from multiple sources into a narrative to tell the story of this understudied topic [18,19]. Key terms and concepts were entered into the databases (ASSIA, Community Care Inform, CINAHL, EBSCO Host, PsychInfo, ProQuest, Social Care Online, Social Policy and Practice, Social Services Abstracts, Scopus, SSRN), along with associated terms (e.g., ‘trans’ OR ‘LGBT’ in addition to ‘adopt’, ‘foster’ OR ‘social work’). Citations were followed up and additional searches were conducted through university library databases, Google Scholar, and specific relevant journal sites (e.g., Sexualities, IJTH, GLBT family studies).

Firstly, we look to the wider research base on the increased recognition and visibility of LGBTQ+-headed families which can be seen to reflect a change in societal constructions of family [20]. A plethora of research has documented the increased options provided to LGBTQ+ (particularly trans) families by advancements in assisted reproductive technologies, including conception via implantation using gametes from one or multiple parents, surrogacy using donated or parental eggs/sperm, and the freezing of eggs/sperm before gender affirmative treatment [21,22,23,24,25].

A dramatic global shift in legal and socio-political climates has arguably encouraged positive attitudes toward diverse families to flourish within certain sects of the public [26,27]. However, discriminatory and exclusionary views and practices are still being reported within numerous sites of family and social life [12,28,29].

Although contemporary studies have begun to attend to trans voices within LGBTQ+ research, lesbian and gay-headed families remain the central focus. Research and policy relating to trans parenting (e.g., via fertility treatments) have lagged behind those supporting reproductive choice for lesbian and gay parents [23,24].

There is a well-established research base focusing on the ways in which heteronormativity (the positioning of heterosexual identities as the norm and all other sexual identities as deviant) disadvantages lesbian and gay adoptive and foster carers, with some attention to the impact of gender-normative processes [3,4,5,30,31,32,33,34]. A raft of research in the 2000s was useful in disconfirming previously held heterosexist stereotypes and beliefs about lesbian and gay parenting disadvantaging children [35,36,37] before later research explored advantages that lesbian and gay families can offer [5,38]. This research was conducted against a backdrop of broader troubling of traditional notions of family, examining same-sex intimacies and deconstructing traditionally gendered family roles [39,40,41]. However, LGBTQ+-headed families still persistently lack the routine acceptance that hetero- and gender-normative families are readily afforded [42,43,44]. The notion of a nuclear family as headed by two cisgender heterosexual parents is given precedence in law as the default model in the United Kingdom [44] as well as in other Western countries such as Australia [45] and the United States [46]. Although calls have been made for research to focus on building a knowledge base that will further academic understanding of trans lives and relationships [47], research on trans carers has been slower to follow [48].

Several papers have explored some element of trans people’s experiences in relation to fostering and adoption. In the United States, Goldberg et al.’s [34] study found reports of invasive questioning about genitals, whether they would make their children trans, and difficulties in matching for three trans participants. Goldberg et al.’s [49] study focused on the findings related to 448 trans people in the United States who were open to adoption or fostering, comparing them to a larger proportion of cisgender sexual minority men and women included within the wider study. Trans participants reported greater fear of discrimination based on gender identity and expression, finance, and social support as compared to their cisgender counterparts. They were more open to placements with ‘harder to place children’, a term used to refer to children that have historically been overrepresented in the equivalent of the United Kingdom’s ‘looked after’ population such as older children, with mental or physical health diagnoses, behavioural challenges, or those from minoritised ethnic backgrounds. Over half the participants reported an identity other than binary trans man or woman, however, findings in specific relation to non-binary identity and potential detriment were not extricated.

In the United Kingdom, Tasker and Gato [50] conducted a focus group study that included six trans and five non-binary people and examined the psychology of future thinking about parenthood. Employing Life Course Theory, they found participants were balancing a desire to parent with other life goals and that some participants did not feel they would fit within the cisgender adoption, fostering, or fertility systems. Though the study did not extricate the views of non-binary people or specifically consider their experience of social work, it highlighted that non-binary participants felt a particular challenge would be recognition of their gender-fluid or neutral parenting intentions. Bower-Brown and Zadeh [51] conducted an intersectional analysis of trans and non-binary (*n* = 7) people’s experiences of accessing parental support spaces and found they experienced some challenges parenting within a normative world, particularly concerning pregnancy spaces. Bower-Brown [52] went on to further examine the difficulties that trans and non-binary people felt when ‘doing parenting’ beyond cisgendered norms, with a particular focus again on biological assumptions made about pregnancy and birthing. Adoptive parents reported an issue in being named ‘mum’ or ‘dad’. Notwithstanding, parenting beyond the binary could be viewed as a strength of trans-headed households [53,54].

Following Hudson-Sharp’s [55] scoping review highlighting a patchy awareness of trans issues in child and family social workers and a systematic literature review identifying the lack of research focus on trans parenting within social work, [29,56] interviewed twenty-five professional stakeholders within UK social work organisations and conducted a thematic analysis of the results. They found evidence of the ability to recognise where stigma was acting upon everyday parenting challenges and a desire for positive change alongside barriers such as a lack of awareness of trans issues in social work staff, lack of confidence in dealing with trans issues, an absence of adequate training in gender diversity, and an absence of a coherent government framework for supporting trans inclusion. Within this study, only one case of a trans person applying to adopt was relayed by a social worker; however, the results indicate a need for further specific examination of social work perspectives on adoption, fostering, and non-binary identity.

Valentine [57] highlights a lack of research examining the experiences of non-binary people as a distinct group, as well as a pressing need to address the Ministry of Justice’s claim that no specific detriment to non-binary people exists that could warrant a review of legislation. Indeed, there is an amassing body of research evidencing that a specific detriment does exist for non-binary people within the UK [57,58]. Matsuno and Budge’s [58] literature review found that non-binary people experience higher levels of depression, anxiety, and suicidality than binary trans people. Non-binary people were found to experience exclusion from services as non-binary identities were not understood or represented [57]. Stonewall’s [59] ‘Vision for change’ for trans people argues for changes to be made to ensure that service provision is offered in an inclusive way.

Research examining non-binary parents and carers’ experiences through a lens of cisgenderism is in its infancy [28,48]. As such, the present study contributes to this emerging knowledge base and attends to the particular dearth of research focusing on non-binary carers’ experiences of cisgenderism in UK adoption and fostering social work practice. This study thus aims to examine whether a specific detriment exists for non-binary adopters and foster carers and to unearth how cisgenderism can affect non-binary carers within UK adoption and fostering social work practice. It will do so by directly accessing the views of non-binary carers and social workers with experience in assessing and supporting them.

## 3. Methodology

In-depth, semi-structured interviews were conducted to explore the subjective experiences of people who self-identify as non-binary and have an interest in or experience of adoption and fostering, as well as social workers with experience supporting non-binary carers.

A subset of data was drawn from a wider UK-based narrative inquiry to address the following research questions:Does a specific detriment exist for non-binary people hoping to adopt or foster in the UK?How can cisgenderism affect non-binary carers within UK adoption and fostering social work practice?

Narrative inquiry can be defined as ‘*the study of the ways humans experience the world*’ (p. 2, [60]) and ‘*a way of thinking about experience*’ (p. 477, [61]). Narrative inquiry is the most appropriate choice for the study because it seeks to explore the co-created expression of a social topic [62].

### 3.1. Procedures and Data Collection

Participants were provided with study information, supported to make an informed choice to take part, had their right to withdraw explained, and provided with written consent. Email or face-to-face interviews were conducted at a venue suited to participant preferences, in accordance with procedures set out in the University of Sheffield’s Research Ethics Committee approval. Interviews were conducted between April 2017 and April 2020 using a topic guide to stimulate discussion regarding their experiences and perspectives on adoption and fostering and the impact of supportive factors and barriers. Riessman and Quinney (p. 393, [63]) assert that narrative inquiry is not an approach that requires a rigid structure or tight boundaries, and instead describe its ‘*essential ingredients*’.

Interviews lasted between 30 and 90 min in-person, or back and forth within one month via email, with questions being sent and responded to at the participant’s pace. Participant confidentiality and anonymity were protected by removing identifying information from all transcripts prior to secure storage. Credibility, dependability, and confirmability of the results were prioritised by methods such as member-checking, whereby information was summarised through the interview to check understanding, and the option to review/edit transcripts and feedback on earlier drafts of analytical interpretation was offered to participants.

### 3.2. Participants

Participants were recruited via purposive and snowballing sampling methods to include a range of identities, increasing the transferability of the findings. Fliers were sent via email to community groups and centres, social media, and research network advertisements, however, it is believed that the sample size was limited in size and to predominantly prospective carers due to the barriers that non-binary people are subject to. People were eligible to take part if they were residents in the United Kingdom, over twenty-one years of age, and had adopted/fostered, applied/planned to adopt or foster or supported trans carers. Six participants were included, representing a variety of genders, ages, and socioeconomic backgrounds, living across urban and rural areas (see Table 1 and Table 2). Targeted recruitment was undertaken to try and recruit participants from minority ethnic backgrounds, however, diversity within the study was limited in this respect. Other identifying information has been removed to protect the participant’s anonymity.

### 3.3. Data Analysis

An inductive thematic analysis was employed using Braun and Clarke’s [1] six-stage method: (1) Familiarisation with the data, (2) Coding (by hand and using NVivo 12), (3) Searching for themes, (4) Reviewing themes, (5) Defining and naming themes, and (6) Writing up. Analytical rigour was ensured by adopting a flexible approach that involved a continual process of comparison and critique of the themes, incorporating reflexivity [64,65]. The essence of each theme, its capability to illustrate the wider issues, and its relation to the telling of the story overall were considered as part of a process of co-creating a narrative representative of the conversations undertaken throughout the interviews.

### 3.4. Researcher Reflexivity

A core aspect of ensuring trustworthiness, credibility, dependability, and confirmability in qualitative research is a recognition of the part that researchers themselves play in the conducting of research. Here, the researcher explored the impact of her identity as a cisgender woman, biological parent, and social worker through an ongoing recording and referring back to thoughts on the interview process, context, participant narratives, and relationships created during the interviews. A model of critical ethical reflexivity [66] was used to address issues of efficacy and ethics related to cis researchers investigating trans issues, enhancing reflexivity and methodological rigour. Further, participants were invited to check and comment upon transcripts and earlier interpretations of findings.

### 3.5. Theoretical Framework

The analysis utilised a framework integrating elements of cisgenderism, stigma theory, and Foucauldian discourses of power to make sense of participant narratives using a broad understanding of gender normativity and associated discrimination [67,68,69]. This conceptual framework was employed as it enabled a detailed understanding of how discourses of power affect non-binary people’s experiences of family. The strategies participants employ to make sense of their non-binary identities are important to unpack to explore modes of acquiescence and resistance. Using a lens that integrated a specific focus on cisgenderism with a wider perspective on stigma and discourses of power sensitised the analysis to the different impacts of each aspect of a person’s identity. As directed by participant narratives, being non-binary (or LGBTQ+) was not assumed to be the only or most important aspect of a person’s identity as a carer/prospective carer. Rather, provision was made to weave a wider consideration of the intersectionality of experience into the overarching analytical framework [70].

Conducting analysis utilising a lens of cisgenderism interweaving stigma theory and Foucauldian ideas of discourse was relevant as cisgenderism is an example of hegemonic discourse, with cisgender stereotypes directing how we see and talk about gender within society. Man/masculine and woman/feminine are largely believed to be the two possible categories of identification, with little opportunity afforded to refute the rigidity of these categories [71]. These binarily gendered discourses do not merely describe the world, they serve to reproduce how it is seen and experienced [68]. Dominant gender discourse seeks to fit everyone into one of the categories and does not make provision for gender variance [17], thus ‘outliers’ become stigmatised. Those who ‘pass’ may be afforded greater societal acceptance; however, non-binary trans people may not align with this idea of ‘passing’ and may be more likely to resist and visibly transgress entrenched binary male-female norms [72]. Further, historical discourse on trans identity has positioned it as psychopathological [71]. Professional discourses (e.g., within social work) then reflect societal power relations influenced by this historical pathologisation of non-cis identities and are enacted by gatekeepers at different levels of adoption and fostering processes [68].

While Foucauldian analysis has been critiqued as being located within a specific time period and for taking an andro- and Eurocentric perspective [73,74] and traditional Goffmanian stigma theory cited as being ahistorical and apolitical [75,76], the application of contemporary theories of cisgenderism and stigma arguably ameliorates these limitations. A critical exploration of relations of power that highlights the disconnect between social work commitments such as anti-oppressive practice and empowerment and its original humanitarian moorings is deemed essential in exposing the shortcomings of contemporary practice [77].

A ‘looking up’ to elucidate how stigma has been designed and activated by state-led campaigns that filter stigma down into everyday interactions was considered important in this study’s analysis. Indeed, Tyler and Slater [75] would argue that a micro-sociological and psychological method of analysing the data could detract from important questions regarding where stigma is produced, how it is produced, and why. This study’s approach thus incorporates micro-, meso-, and macro-level analysis of individual, organisational, and societal level stigma, to provide a fuller picture of the stigmatising impact that dominant cisgenderist societal discourse can have on non-binary parents and carers [10,68,69,75,78].

## 4. Integrated Findings and Discussion

### 4.1. Theme 1: Barriers for Non-Binary Carers

Non-binary participants believed numerous barriers existed that would prevent them from starting or growing their family via adoption or fostering. For example, Jamie believed being that being non-binary could cause such significant detriment that they may need to try and hide their identity to be approved as an adopter:

I was really concerned about, like, being a queer family… if we were a cis gay couple, I’d be like, it’s pretty mainstream now, I think we’re fine, you know. But… am I gonna have to pretend to be [cis woman]… like, could I even do that? Are we gonna have to pretend to be a straight couple? (Jamie, non-binary prospective adopter)

When considering presenting themselves publicly as part of an adoption assessment, a context in which they could be disadvantaged by being non-binary, Jamie and other participants believed that they may need to ‘pass’ to be successful. Jamie reported a pressure to perform a cis identity in public and present to an adoption agency as female to avoid the social sanctions they believed would be placed on them and their husband. They believed they would not be accepted as a valid form of parent if they did not align with socially protracted cis- and heteronormative ideals of a man–woman relationship [79] which have been transmuted onto ideals of how families should be headed [80].

Some participants asserted that they would not hide their non-binary identity, however, there remained a reluctance to offer it openly:

I don’t think I would deny it. But I wouldn’t be the first to bring it up… (Ash, neutrois prospective adopter)

Still, there was a resistance to openly displaying one’s true gender identity, indicating a belief that being non-binary would denigrate Ash’s position in terms of how they would be viewed by an adoption agency. Valentine [57] similarly found that non-binary people did not feel comfortable sharing their gender identities with services (except for specific LGBTQ+ services), expressing little confidence that health and social care services would respect their identities. Valentine [57] found that while using services within the previous 5 years, 67% of non-binary people had been misgendered by accident, 33% on purpose, and 49% felt like they were educating professionals.

This finding offers support to Bradford and Syed’s [72] argument that those with transnormative presentations are more likely to feel included in social life. Those who ‘pass’ are seen as acceptable parental figures as they maintain societal binary gender structures. Whereas those whose identities transgress gender norms present a threat to social order. Non-binary identities resist the established gender norms that are upheld within society and bound to family life. Here, stigma acts as both a productive and constitutive force [10], enabling power to function and dictate who is regarded as a good parent/carer and who is kept out of adoption and fostering systems via gatekeepers. 

A similar process priorly restricted lesbian and gay carers [3,4]. The belief was that the further a person’s gender expression sat from established binary gender norms, the more difficulty they would have in adopting:

Can you imagine someone has a sort of more queered presentation?... They would have a tougher time than me…I think they’d really struggle. (Celyn, non-binary adopter)

Again, although Celyn was ‘out’ in their adoption process, they still felt that the capacity for them to be read in a normative way aided their adoption journey. The implication being that the further you depart from the rigidly established gender binary present within society, the more difficulty you will have in being accepted throughout your interactions with the social work system. Indeed, a direct comparison was drawn whereby it was felt that being a non-binary as opposed to a trans person with a binary identity would present markedly increased difficulty for those being assessed as adopters:

You can be in Annie’s position [*Ash’s wife, a woman with a trans history*] and get away with it… Because she’s binary. And so, she can just say ‘I am a woman’, and not have to qualify it. And people will accept that. (Ash, neutrois prospective adopter)

Here, Ash asserts that their wife Annie could move through social work systems without having to explain her gender identity. Because she aligns with binary expectations, she does not have to explain herself. To the contrary, Ash feels they would not be accepted as they do not fit with the male or female categorical groupings that society at large, and as such, social work, uses to process information on a person. Social workers hold a substantial amount of power in relation to the approval of adoptive families because it is their interpretation of a range of interactions that directs decisions [81]. The powerful influences of gender normativity can mean that the public performativity of gender is different from that of the personal [82]. Processes of normative social categorisation can result in individuals portraying identities considered by the majority to be more typical, to either feel a sense of belonging or to give the agent a form of political power. In the context of adoption and fostering, the form of power that may be needed by an agent is that of control over their personal information.

The result of gender-normative views being embedded within society and social work practice is that non-binary identities are stigmatised, spoiled, and subjugated by prevailing discourses of power directing the majority to view them as being socially undesirable [67,68,69].

I never thought it would be as difficult… when we went to panel and they got approved, I actually felt so confident about Charlie. I just thought, they’re amazing…‘they’re gonna get snapped up’… It didn’t happen… it was nearly a year…once we got into family finding it was almost like we were back in time, when, when, er, LGB, you know, lesbian and gay adopters were seen as n-, as just, as not good as heterosexual. (Amy, social worker)

These quotes indicate that the hierarchy of gender and sexualities [83,84] has shifted within adoption and fostering practice. Where gay and lesbian adopters and carers were previously regarded as viable carers alongside a prioritisation of heterosexual couples as the ideal [3,4,6], trans and particularly non-binary adopters/carers are now perceived to be placed at the bottom of the hierarchy [28]. Participants believed they would need to conceal a lack of adherence to gender-normative presentations to prevent discrimination at the micro-level from individual social workers and at the meso-level where non-binary gender identity could prevent them from gaining approval as carers from adoption or fostering organisations. Macro-level societal negative perception of non-binary people is additionally indicated as social work perceptions of the ideal family are derived from societal perceptions of this ideal.

### 4.2. Theme 2: Prejudice in Adoption and Fostering Matching Processes

The social workers interviewed reported both overt and covert gender discrimination in the processes of matching non-binary carers to adoptive and foster children. For this theme, the social workers’ voices are focal because within the UK system (with the exception of profiling events), much communication supporting the marching part of the process is done ‘behind the scenes’ on behalf of the adopter, not by them.

I don’t think that people were like, overtly prejudiced…they would say well how do, how would they respond, how would they explain their transgender status to a child? A child might get confused… how would a child er, cope with bullying? (Amy, social worker)

Social workers like Amy have a dilemma in terms of what guides the decisions they make to serve a child’s best interests, seemingly influenced by moral ideals about how to best raise a child that have been transmitted culturally [85]. Here, social workers were attempting to make best interests decisions guided by cisgenderist notions [69]. Presuming that a child would not be able to understand a parent’s non-binary gender arguably reflects the macro-level cisgenderist discourse that has infused their micro-level personal and professional thinking about gender and as such influences the meso-level organisational level actions taken with regards to approval and matching. Research shows that children understand their gender from an early age [86], corroborated by this study’s participants such as Jamie who reported knowing they were not a girl or boy by 2–3 years of age. A further cisgenderist assumption is that a child placed with a non-binary parent would be bullied and that a child placed with cis parents would not experience bullying. However, a recent review of bullying suggests that its reasons are complex, encompassing individual and contextual factors such as the school environment and whether families encourage their children to solve problems using violence [87].

Amy reported examples of overt transphobia being expressed by other social workers, relaying a comment made by a social worker at a potential placing authority after a meeting with a non-binary adopter:

‘I just think, they’re confused about their gender, their identity’. (Amy, social worker)

Denying the validity of a non-binary carer’s identity demonstrates the transphobia that exists in contemporary society [16] and is transmuted via discourses into social work practice [68]. Within a normative ideological construction of gender, cisgender identities are positioned as natural, fixed, and indisputable, with non-binary identities placed as unnatural, deviant, or other [88]. Reports indeed indicated that being non-binary was something that social workers were unfamiliar with, and thus regarded as problematic:

I think the non-binary aspect of trans confused people [social workers considering a match with an approved non-binary adopter] even more if I’m honest… it was a bit feeling it’s unknown, we’ve got nothing to base it against. (Melanie, social worker)

Gender identities, expression, and related terminology that sit outside of dominant gender norms [69] were met with anxiety and confusion from adoption and fostering social work professionals. Indeed, previous research has found that many child and family social workers had not taken part in gender diversity training, and the knowledge found across practitioners varied with only pockets of gender awareness [55]. This lack of knowledge could result in a potential match not being explored:

So, you’d go to a profiling event *[where adopters and their social workers meet with family-finding social workers]* and people would have conversations with Alex, and Alex looks female… and those conversations would be really positive… we’d have conversations afterwards and share stuff, and then it would just go quiet… they would say ‘that’s not a problem’ [referring to Alex being non-binary], and then you wouldn’t hear anything. (Melanie, social worker)

However, this example may demonstrate an issue that goes beyond lack of confidence; it could evidence covert (hidden or unconscious) discrimination, which can be more pervasive [89]. Social workers are influenced by gender-normative discourse that places cis adopters and carers as the only valid categories of parent, closing off conversations regarding non-normative options [17,68]. Challenging decisions that social workers, carers, or adopters feel to be discriminatory but have no explicit, recorded proof of would be incredibly difficult. To make a case that the Equality Act 2010 [90] or Gender Recognition Act 2004 [91] has been contravened, evidence must be presented to a tribunal or court. However, where gender is not overtly stated by workers as a reason for not selecting a trans person as a suitable match for a child, it cannot be identified as a discriminatory decision nor formally challenged as such.

Some outright discrimination and some… trying to hide behind other things… agencies would come up with lots of different reasons, none of which really made sense… That was easier than to say, ‘we’re struggling with this’. (Melanie, social worker)

Melanie’s quote suggests that social workers verbalised another reason (than gender) to justify not exploring a match between a child and a non-binary carer, exemplifying micro- and perhaps macro-level cisgenderist views. Between them, Melanie and Amy reported having interactions with a large proportion of the local authorities spanning the United Kingdom, via online and in-person linking and matching methods. As such, their comments offer a transferable insight into the prevalence of micro- and meso-level overt and covert discrimination within adoption and fostering in the United Kingdom. Although overt discrimination can be challenged under legislation such as the Equality Act 2010 [90] and, indeed, Melanie expressed being motivated to challenge this, covert discrimination can be more pervasive and problematic to address [89]. Resultingly, a reluctant acquiescence occurred within the professional participants as they could not see a clear way to challenge cisgenderist professional encounters that were masked with ambiguity.

Misgendering, a form of pathologising [92], was another form of gender-based discrimination found to be prevalent within social services:

It must be so difficult for them, you know, they constantly…they get misgendered… all the time. (Amy, social worker)

Social work participants believed discrimination due to being non-binary was occurring, but it was difficult to isolate the process by which negative responses were due to carers being non-binary and not some other characteristic in order to complain:

My experience of family-finding was that it was just a ‘no’ or a ‘we’re not starting a discussion’. And there was never really any opportunity to try and talk about Alex. (Melanie, social worker)

Social workers used their agency to attempt to shield adopters from examples of direct transphobia:

There were some things where I, I wanted to talk to Charlie [a prospective adopter who identifies as non-binary] about the, the sort of prejudice and discrimination, but also didn’t want them to realise how sometimes, how difficult it was… because it just seemed too, too, too much. (Amy, social worker)

Many of the impacts, such as minority stigma stress, mental illness, and isolation, are known to be experienced by people following exposure to discrimination [93]. It would indeed be the role of a social worker to advocate for an adopter/carer and challenge discrimination based on their gender identity, supported by legislation. However, the protection that the law claims to afford social work service users [94] has been shown by this study’s findings to be insufficient for adoption and fostering work with non-binary people.

### 4.3. Theme 3: Intersectional Disadvantage

Participants who were non-binary and had intersectional experiences of also being disabled or experiencing mental illness believed that they would experience pronounced specific detriment in their aim to become parents/carers. Cho, Crenshaw, and McCall [70] argue that there is potential for greater threat to a person’s well-being where they experience discrimination in relation to multiple dimensions of their identity:

There is a ‘you can adopt whether you’re…’ and there’s a list of identities, but there’s never anything disability specific… [Ash read about] the first person in the UK to adopt as a solo parent who required full-time [personal assistant] care…And they really struggled with questions about their capability and about whether it would be someone else doing all the parenting? Just in the assumption that he couldn’t possibly parent. (Ash, neutrois prospective adopter)

Ash expresses a concern that it has not been made plain or visible that disabled people can adopt. Their expectations about whether or not they will be able to adopt as a disabled person have come from negative media coverage, implying that disabled people are unable to parent, and that parenting itself can and should be distilled into physical manifestations of care, as opposed to considering the multifaceted support that a parent gives a child, rooted in the emotional and psychological realm.

Clarke and O’Dell [95] explore how discourse on disability has placed disabled parents as being dependent on and looked after by their children. This positioning of disabled parents has been critiqued as marginalising disabled parents; however, the legacy of the discourse remains [68]. A social model of disability would offer an alternative analysis of Ash as being prohibited from fully taking part in the family life they choose because of barriers that society constructs [96]. Indeed, as Ash’s identity lies at intersections of gender, sexuality, health, and disability, they experience a complex web of societal disadvantage [97]. Within this web, cisgenderism is enmeshed and inextricable, affected by and affecting other dimensions of difference, serving to exacerbate processes of ‘othering’.

Within an adoption or fostering assessment, the stability of an applicant’s mental health is assessed [98]:

Parts of the assessment that address mental health would be difficult for me to talk about. I have been to the doctors a few times and I have phrased it as “Low Mood” and played it down a lot due to fear that my poor mental health will go on my medical record and be held against me in the assessment process. I got the impression from a meeting I went to about this for X Fostering and Adoption that they aren’t very forgiving when it comes to mental health, so if you have experienced any kind of depression, self-harm or suicidal thoughts in your past this is held against you. (Toby, non-binary prospective foster carer)

Toby similarly considers that their intersectional experience of being non-binary and a period of poor mental health could interact to disadvantage them in a way that isn’t ‘very forgiving’. The way they describe being unwell as being ‘held against you’ suggests that mental health status could be used alongside gender status as a gatekeeping tool to keep those who are not seen as ideal carers from entering the system of assessment.

For trans and non-binary people, mental illness and self-harm may be more common when they experience discrimination in society [16,98]. As such, to expect a prospective adopter or carer not to have encountered these issues is a hidden form of cisgenderist discrimination. Much like a need to hide gender identity then, non-binary people may feel a need to minimise their experience of mental illness to be approved as adopters or foster carers. However, it has also been found that mental wellbeing is improved when young people receive gender affirmative support from a young age [86].

The current adoption and fostering assessment guidance lack consideration of the differing experiences that non-binary people have as it relates to a gender-normative majority [98,99]. Indeed, Barksy [100] argues that assessment methods are outdated as they have been designed based on the assumptions that being cisgender and heterosexual are the prevailing and preferred forms of gender and sexuality. Instead, he offers a modified and more inclusive genogram model for social workers to use. However, while this is helpful in recording different familial and support networks, it does not account for the minority stress and resultant mental health experiences that trans people have [93]. The integration of a structural understanding of the impact of stigma operating at a societal level could reframe mental illness in trans people as an expected and normal human reaction to the extreme emotional pressure that cisgenderism exerts on non-binary people.

## 5. Conclusions and Recommendations

This study found evidence of overtly and covertly operating cisgenderism that disadvantages non-binary people within UK adoption and fostering. Jamie and Ash worried that they may need to ‘pass’ to be approved, and Celyn believed that if they were ‘more queered’ in presentation they may have struggled more to be approved. The fear of bias expressed by non-binary people shows that people with non-binary identities can face an even greater disadvantage than those with binary trans identities, as they fail to acquiesce by ‘passing’.

Amy and Melanie recounted how other professionals struggled with the idea of non-binary gender identity, reporting misgendering, misunderstanding, and worries that children placed with non-binary carers would be confused by their carers’ identities. Overt discrimination was found to be less prevalent; however, specific instances of covert or unconscious bias and discrimination were common within many of the matching/family finding interactions. Hidden or unconscious, covert discrimination can be more pervasive [89] and there exists no direction in the relevant legislation [90,91] regarding how hidden discrimination or unconscious bias in relation to non-binary people can be addressed.

Toby and Ash reported that their intersectional experiences of having a disability or a period of poor mental health can further exacerbate the specific detriment they may face. Looking up at the forces that shape the emergence of stigma in adoption and fostering contexts is an essential aspect of addressing the complexity of interacting factors of bias [75,78]. Adoption and fostering services may be mirroring trends in mainstream media, whereby increased rights and support are given to some trans communities while other communities remain marginalised. State-sanctioned stigma is produced (e.g., politicians, journalists, and TV producers) to craft an image of non-binary people as the deviant ‘other’, to maintain a status quo of gendered roles and power divisions in sites of family (among other aspects of social life) [101]. When discrimination is enshrined in public institutions, discriminatory actions are effectively authorised. Here biopower acts to manage the population through administrative governance of living beings [68]. Stigmatisation is enmeshed in any project of neoliberal governance [101] as actors within that system of governance direct societal views of what constitutes a ‘good family’, positioning cis-headed families as normal and desirable, and non-binary-headed families as deviant.

This study develops previous knowledge on trans and non-binary people’s experiences of adoption and fostering [28,34], an earlier conference paper presented at the 6th International Conference on Gender Research [102], practitioner experiences of trans parents [56], and trans people’s parenting desires [50]. It contributes to the literature by adding a novel perspective that directly accesses the experiences of non-binary people and their supporting social workers in relation to specific detriment within UK adoption and fostering systems. The findings add to a growing body of research suggesting that non-binary people in the United Kingdom do experience a specific detriment that necessitates a review of legislation, policy, and procedure [57,58]. While it is located within the adoption and fostering social work sphere, the results of this exploratory study can be applied to a range of family contexts and have direct application for professionals seeking to understand the detriment faced by non-binary families. It is however highlighted that future research is needed to explicate the complexity of the social processes that cause specific detriment experienced by non-binary carers, as well as an international exploration of this issue.

To conclude, findings highlight how embedded gender norms, particularly cisgenderist discourses, influence how people can understand and express their gender identities within society. Identities that sit outside of the accepted gender norms of the time are limited, marginalised, devalued, and invalidated. The resultant effect is that non-binary people experience multiple challenges as they try and navigate their family lives in a society that is organised around binary gender identities.

## Figures and Tables

**Table 1 behavsci-14-00614-t001:** Interviewee characteristics: non-binary-identified participants.

	Pseudonym	Gender Identity	Age Bracket	Interview Date and Type	Other Personal Characteristics
1	Celyn	Non-binary	25–30	17 July 2018 In person	Bisexual, queer, single. White Welsh. Adopter with a child in placement who is gender questioning and has behavioural concerns.
2	Jamie	Non-binary, Trans/trans- masculine	25–30	12 January 2019 In person	Prospective adopter. Pansexual, married to cis man. White American, lives in an urban area in the south of England. Has an invisible disability.
3	Ash	Neutrois	25–30	14 January 2019 In person	Prospective adopter. White British, north of England. Pansexual, married to a woman with trans history. Biological son with previous partner. Disabled, has a personal assistant.
4	Toby	Non-binary trans man	25–30	20 January 2020–29 January 2020 Email	Prospective foster carer. White British, from a religious family in South of England. Pansexual, monogamous, single.

**Table 2 behavsci-14-00614-t002:** Interviewee characteristics: social workers.

	Pseudonym	Professional Role	Age	Date Interviewed	Profile
1	Melanie	Adoption social worker	30–40	17 July 2018 In person	British, minority ethnic background, cisgender, social worker. Adoption 1-year, experience supporting non-binary carers, previous child protection role.
2	Amy	Senior social worker: adoption	30–40	17 July 2018 In person	White British, heterosexual, cisgender, social worker. Worked in adoption since 2009 (local authority), moved to voluntary adoption agency in 2015.

## Data Availability

Data reported within this study are not publicly archived.

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
