# Peer review of "‘Are We Gonna Have to Pretend to Be a Straight Couple?’: Examining the Specific Detriment that Cisgenderism Places on Non-Binary Adoption and Fostering Applicants in the United Kingdom"

_behavsci, 2024, doi:10.3390/bs14070614_

Round 1

Reviewer 1 Report

Comments and Suggestions for Authors

Please see attached for my complete review letter. I have also pasted the letter below, but the formatting may be difficult to follow:

Thank you for the opportunity to review this manuscript. This is an important topic and I commend the authors for their efforts to find out more about how the children’s social work system interacts with non-binary parents. The authors do a particularly sensitive job to naming the risks and benefits to further spotlighting vulnerable transgender and non-binary people and gives strong rationale for the usefulness of this work.

This article has a robust literature review that seems well formulated given the current state of the literature. In a few places below, I give suggestions of specific additional papers that could be included to strengthen specific points or give more recent examples. I also recommend adding some discussion of the current literature around trans and non-binary parenting experiences and strengths if space allows.

The authors are clearly knowledgeable about this area and the data is interesting and useful. The theory is well explained and integrated.

Overall, the manuscript has the possibility to contribute meaningfully to the existing literature and to social work practice. However, there are several things that could be addressed to strengthen the work and prepare it for publication:

1.     The sample is very small—only four non-binary people, only one of who has started or experienced the foster care or adoption process. The other three are prospective adopters or carers. There are also two cisgender social workers familiar with working with non-binary carers.

a.     If this is the complete sample and more data cannot be obtained, the title and framing should acknowledge that this is mostly a sample of non-binary prospective carers, and not current carers or parents. The findings are still useful, but they are speaking more (or at least as much) to the barriers to pursuing parenthood through this method as they are detriments in the system once engaged.

b.     Further justification is needed for the small sample and the limited experiences of the sample to the topic

2.     Participants are not sufficient anonymized to protect their identities. Authors should explain if the participants approved this level of specificity in their descriptions. Attention should be made to decreasing the specificity of the descriptions where the detail is not necessary. Further, particular attention must be paid when describing a minor child in a way that could identify them.

3.     The findings are communicated in conjunction with the discussion in a way that I believe takes away from understanding the data and voices of a marginalized group. I believe that separating the findings and discussion, attending to giving enough voice to the participants, and more thoroughly explaining the themes is necessary. It seems that there are deeper and richer themes in the data than are named, but the author is explaining them through pulling in outside voices, rather than shining a light on what is in the (very interesting) data! It is particularly notable that there are very few quotes from the non-binary participants, while the social workers have more space for their voices. There are more non-binary participants, and they are the focus of the article. It would be a stronger manuscript and analysis if those voices were to shine through. At the end of reading this, I find that I cannot imagine what the conversations were like with the non-binary people or what their major thoughts and concerns were.

a.     Overall, it seems that there is some more analysis to do of the data—what is found here? What are the participants truly saying and what emerges from the data? When that is firm, it seems that there would be less need to explain the themes with so much outside content. As it is written, it gives the impression of hunting for a quote that illustrates what the literature already says or theory already suggests, rather than coming from the data.

b.     However, I acknowledge that this may be an epistemological choice on the part of the authors. If they are aiming to present a co-construction of the data with the participants and the authors contributing, that should be further explained and justified in the methods.

4.     The description of the methods is overly long without providing much novel content. The authors explain in detail the steps of the seminal Braun & Clarke (2006) article, without giving much detail about what is unique and strong about their application of it. I would recommend a combination of shortening unnecessary elements and expanding on the richness of specificity to this data and the theoretical framework.

5.     I might recommend this very helpful article that can help to address #3 and #4:

a.     Goldberg, A. E., & Allen, K. R. (2015). Communicating qualitative research: Some practical guideposts for scholars. Journal of Marriage and Family77(1), 3-22.

Below, please find my line-by-line feedback. I enjoyed reading this manuscript and believe it has potential value to the field. I am grateful to the authors for working to give voice and attention to this important topic.

Comment by line/page:

·       Line 91: The Mallon and other citations—although important—are quite dated. Could consider incorporating this more recent article about heteronormativity in the adoption and foster care space:

o   Goldberg, A. E., Frost, R. L., Miranda, L., & Kahn, E. (2019). LGBTQ individuals' experiences with delays and disruptions in the foster and adoption process. Children and Youth Services Review106, 104466.

·       Line 94: A more recent and decisive set of papers to add to the “raft” of findings about queer parenting outcomes:

o   Farr, R. H., Forssell, S. L., & Patterson, C. J. (2010). Parenting and child development in adoptive families: Does parental sexual orientation matter?. Applied developmental science14(3), 164-178.

o   Farr, R. H. (2017). Does parental sexual orientation matter? A longitudinal follow-up of adoptive families with school-age children. Developmental psychology53(2), 252.

·       Line 141-143: This level of detail about the methods of the study seems unnecessary

·       One area of the literature review seems a bit lacking: summarizing what is currently known about the experiences and strengths of transgender and non-binary parents! Perhaps after discussions of challenges, this could be added? Earlier in the literature review, the authors address the literature about lesbian and gay parenting. The authors also address experiences specific to the foster and adoption process for trans people. There seems to be space to talk specifically about parenting experiences for trans and non-binary people and the evidence of their parenting strengths! Here are a few options:

o   Tornello, S. L., Riskind, R. G., & Babić, A. (2019). Transgender and gender non-binary parents’ pathways to parenthood. Psychology of Sexual Orientation and Gender Diversity6(2), 232.

o   Riskind RG and Tornello SL (2022) “I Think It’s Too Early to Know”: Gender Identity Labels and Gender Expression of Young Children With Nonbinary or Binary Transgender Parents. Front. Psychol. 13:916088. doi: 10.3389/fpsyg.2022.916088

o   Tornello SL (2020) Division of Labor Among Transgender and Gender Non-binary Parents: Association With Individual, Couple, and Children’s Behavioral Outcomes. Front. Psychol. 11:15. doi: 10.3389/fpsyg.2020.00015

o   Worthen, M. G. F., & Herbolsheimer, C. (2022). Parenting Beyond the Binary? An Empirical Test of Norm-Centered Stigma Theory and the Stigmatization of Nonbinary Parents. LGBTQ+ Family: An Interdisciplinary Journal18(5), 429–447. https://doi.org/10.1080/27703371.2022.2123422

Methods:

·       The research questions are not framed as questions. This is a minor point, but it would be more clear if the questions were framed as questions, or if the research aims were named as research aims.

·       Line 190—need details about what an “email” interview would be. The face-to-face interviews are described in detail, but little is explained about how a similar procedure could be conducted over email.

·        Line 189-190 and then 215-216 both described informed consent. This is only needed once

·       Line 210: Six participants is quite a small sample—especially when broken down into four non-binary individuals and two social workers. Further then that only one of these participants actually adopted or fostered. It seems that this group is not particularly representing non-binary adoptive parents and foster carers—but rather perspective ones. If this is the case, the study should be framed and justified this way.

o   I would like to see this small sample size further justified. Is the gap in the literature so large as to justify such a small exploratory study? Does the author have reason to believe that the detriments found discourage non-binary people from pursuing adoption and foster parenting so much that prospective parents are the target study group? Are there not many non-binary foster carers or adoptive parents?

·       Table 1: This level of detail in this table pushes the boundaries of anonymity. Did the participants approve of this level of detail? I would significantly decrease the level of detail to make identification more difficult.

o   For example, a person who knew “Celyn” would likely easily guess it was them based on:

§  Age 27, non-binary, “Bisexual, queer, single. White Welsh, lives in north of England. Adopter with 5-year-old child in placement for less than a year who is dual heritage, gender questioning and has experienced prior abuse that triggers child to parent violence.”

o   How many people matching that description exist? I would think only one.

o   I encourage the authors to inspect what is necessary and generalize what is not:

§  Some examples of my recommendation would be:

§  specific areas are likely not necessary, can you describe urban, suburban, rural? Or if what matters is political nature of the area, can you give a broad description?

§   Specifics of child’s trauma and behaviors are not needed and the child did not consent to their information being shared. Can you drop those details and specifically name gender questioning (seems relevant) and with “behavioral concerns”

§  No need to name specific degree “full-time graduate student” is sufficient

§  No need to name specific number of hours a PA provides care—can just say that receive assistance

·       It is uncommon for someone to break out the six themes of Braun and Clarke’s thematic analysis in writing this way and to describe them in such depth. Further, Braun and Clarke (2019) have warned against the rigid application of the “steps” and encouraged more reflexive and flexible application.

o   Braun, V., & Clarke, V. (2019). Reflecting on reflexive thematic analysis. Qualitative research in sport, exercise and health, 11(4), 589-597.

·       The methods can be more briefly described, with a focus on specific methodological elements that were important to this particular project and with less focus on typical methodology. Also be mindful of when language in the methodology section closely mirrors the language of the Braun and Clarke (2006) paper.

·       Line 271: this reflexivity statement would be strengthened by the author naming positionality in terms of parenting. Are they a parent? An adoptive or foster parent? A person who was fostered or adopted? How might those positions change the analysis? What about positionality related to the broader LGBTQIA community?

·        Theoretical framework: The authors do a thorough job introducing their theoretical frameworks and justifying them. I would like to see the authors emphasize more about how these theories were employed in their methodology and how they specifically sensitized the analysis. This could be expanded on in place of general method description.

o   I am curious why the authors chose a broad stigma theory and not a more specific theory like queer family theory?

o   Allen, S. H., & Mendez, S. N. (2018). Hegemonic heteronormativity: Toward a new era of queer family theory. Journal of Family Theory & Review10(1), 70-86.

·       Line 326: “believed being this could cause such…” --is “this” a typo? Being non-binary?

·       Line 330: “gona” –is this misspelling in context? Or a typo?

·       Line 346: neutrois is a less common term than non-binary. Could you define in a footnote?

·       Line 361: Who is Annie? Need to contextualize what is said.

·       Theme 1 notes: There a powerful quotes in this section and an opportunity to connect and explore them. However, the authors take up most of this section with their own analysis and the voices of other scholars and discussion. It feels as though the reader cannot see or hear what the participants are saying—the lived experiences of the people—through all the interpretation. I would like to see more thick, rich description of what was actually present in the data and what the “meat” of the theme is. The theme is never described beyond “participants believed there would be barriers.”

o   I would potentially like to see the discussion pulled out of the findings—giving space to these marginalized voices. There should also be space for rich description of the theme—there is more than they believed their were barriers! I am reading a consistency in participants’ beliefs that being non-binary is a *particularly* stigmatized identity, over other queer or trans identities, and this pull toward a less queered presentation. And then—the social worker voice CONFIRMS the fears named by the prospective adopters!!! This is great rich data—but feels lost in the intense level of analysis and connection to outside literature.

o   Overall: position the participants as the most important part of this analysis, and the language around their examples of the theme should enrich, explain, and situate the themes. Right now, it reads as the authors taking the points as jumping-off points for their own thoughts and knowledge of other writers.

·       Line 401: is Amy’s statement here theoretical? Or based in experience? I would like to see authors explain and situate quotes more. How is this theme different from what Amy shared in the previous theme? Need explication.

·       Line 408: A great point to connect to research specifically about how non-binary and trans parents do this and how children understand it! Also findings about how children understand their own gender identity from an early age, for example:

o   Hässler, T., Glazier, J. J., & Olson, K. R. (2022). Consistency of gender identity and preferences across time: An exploration among cisgender and transgender children. Developmental Psychology, 58(11), 2184–2196. https://doi.org/10.1037/dev0001419

·       Line 415-419: This is a nice description of what was said, situated in context with a quote—I would like to see more of this! Also, can the authors explain if this was the only person (of the two social workers) that said this? It doesn’t seem accurate to say, “social workers reported” when only one did. While there is debate about the utility of specific counts in reporting themes, we do need to be accurate about who said what.

·       Line 437-442: Explain who the people are at these profiling events, as the types of events vary greatly across regions, countries, etc. Where they talking to prospective parents? Workers? Other foster carers?

·       Line 450-468: This section is such a strong section! I love that I can read the quotes and deeply understand what the workers said, what the context behind it was, the broader over-arching theme, and how it is situated in context of the law. I would love to see more of this type of discussion of the data in the findings with a separate discussion that could more clearly let voices of participants sing without losing the authors’ well-reasoned discussion.

·       Theme 2: I am struck by how much more of a voice and space in the document the two social workers had over the four non-binary participants. I hope this changes for theme 3, but overall, we are hearing much more about what social workers think and very brief, and very few statements from the non-binary people themselves.

·       Line 524-526: This statement can be phrased more accurately to avoid the misunderstanding that trans and non-binary people are mentally ill. Yes, there is a higher incidence of mental health difficulties, self-harm, and suicidal ideation among this population when they have faced discrimination, denial of their identities, and barriers to affirming care. See Hassler et al. (2022) above showing that this is not necessarily the case for people with affirming treatment across their lives. A little more context is important here! The point the authors are making is still an important one!

·       Theme 3 overview: A few more (very exciting!) quotes in this section from the non-binary folks. I would overall expect to hear more from these marginalized and understudied people—rather than mostly relying on the social worker’s voices for the majority of the narrative. Further, as the author is a social worker themselves, it may be a useful point of reflexive practice to think about the urge to jump in and explain or contextualize what the non-binary people are saying, rather than letting them stand on their own words. Is there a privileging (of course, unintentionally) or system-empowered voices over those experiencing the stigma and discrimination?

·       Line 542: Because the conclusion and recommendations are lengthy—I would like to potentially see the discussion material moved to this section as a discussion and conclusion, giving the themes and quotes more space. If the authors are going to keep the findings and discussion combined to this level, I would like to see the methods address more of the co-construction of meaning happening between the participants and the author as they are given equal space—if not the authors given more space—to discuss the experiences and interpretations.

o   Further, the conclusion and recommendations seem to emphasize more of what the author and social workers elaborated on than the specifics of what the non-binary participants shared. What were their major concerns? Can you make sure to include them clearly? Which participants recounted experiences of bias? It is much stronger to explain that non-binary people feared this bias and described situating their fears in lived experience, and reading about others experiences, while social workers could recount specific incidences demonstrating the exact types of treatment the non-binary prospective adopters named! (Rather than saying “participants indicated there was bias.”)

Author Response

Thank you for the wealth of constructive critique points that you have provided. These have been instrumental in helping me improve the manuscript. Please see attached the table I have used to address each reviewer point in turn, and the highlighted sections in red, as per the guidance provided. 

Thank you for kindly taking the time to read and feedback on my work.

Reviewer 2 Report

Comments and Suggestions for Authors

Thank you for allowing me to review this paper.  The paper details a qualitative exploratory study, undertaken in the UK, which examines the detriment experienced by non-binary and trans adopters and/or foster carers and also incorporates the perspectives of two social workers.  There is minimal research published internationally regarding gender minority adopters and/or foster carers.  Indeed it is severely lacking in the UK also. Therefore this paper seeks to add to the sparse body of knowledge.

The manuscript is clear and is well structured.  The themes are appropriate and there is clear links to current and historical research to support and also refute the findings.  The conclusions drawn allow for consideration of impact of this marginalised group within UK social work practice.

There are a few comments that I would like considered, please see below:

Line 8: why are you detailing adopters, parents and foster carers? Adopters are parents.  Therefore do you mean biological parents?

Line 58: term is used within this paper readers make sense of the narratives of non-binary people. This needs clarifying as not certain what is meant.

Section 2 – literature review.  I’m not certain how the review has been conducted and by terming the section ‘literature review’ I expected a clear framework.  I would suggest renaming as ‘Background’.

Line 80: you mention heteronormativity, but haven’t defined it.  I would advise you to insert a small sentence detailing this term.

Line 138: identified should be identifying

Table 2: Adoption social worker and senior practitioner. What’s the difference in the terms?

When determining codes and themes, the use of NVivo is discussed, but did the author sense check with anyone else?

Author Response

Thank you for the feedback points that you have provided. These have been useful in helping me improve the manuscript. Please see attached the table I have used to address each reviewer point in turn, and the highlighted sections in red, as per the guidance provided. 

Thank you for kindly taking the time to read and feedback on my work.
